# Protonic solid-state electrochemical synapse for physical neural networks

Xiahui Yao [1], Konstantin Klyukin[2], Wenjie Lu[3], Murat Onen[3], Seungchan Ryu[4], Dongha Kim [2], Nicolas Emond [2], Iradwikanari Waluyo[5], Adrian Hunt[5], Jesús A. del Alamo [3✉], Ju Li [1,2✉] & Bilge Yildiz [1,2✉]

Physical neural networks made of analog resistive switching processors are promising platforms for analog computing. State-of-the-art resistive switches rely on either conductive filament formation or phase change. These processes suffer from poor reproducibility or high energy consumption, respectively. Herein, we demonstrate the behavior of an alternative synapse design that relies on a deterministic charge-controlled mechanism, modulated electrochemically in solid-state. The device operates by shuffling the smallest cation, the proton, in a three-terminal configuration. It has a channel of active material, $WO_3$. A solid proton reservoir layer, $PdH_x$, also serves as the gate terminal. A proton conducting solid electrolyte separates the channel and the reservoir. By protonation/deprotonation, we modulate the electronic conductivity of the channel over seven orders of magnitude, obtaining a continuum of resistance states. Proton intercalation increases the electronic conductivity of $WO_3$ by increasing both the carrier density and mobility. This switching mechanism offers low energy dissipation, good reversibility, and high symmetry in programming.

[1] Department of Nuclear Science and Engineering, Massachusetts Institute of Technology, 77 Massachusetts Avenue, Cambridge, MA 02139, USA.
[2] Department of Materials Science and Engineering, Massachusetts Institute of Technology, 77 Massachusetts Avenue, Cambridge, MA 02139, USA.
[3] Department of Electrical Engineering and Computer Science, Massachusetts Institute of Technology, 77 Massachusetts Avenue, Cambridge, MA 02139, USA. [4] Department of Mechanical Engineering, Massachusetts Institute of Technology, 77 Massachusetts Avenue, Cambridge, MA 02139, USA. [5] National Synchrotron Light Source II, Brookhaven National Laboratory, Upton, NY 11973, USA. ✉email: alamo@mit.edu; liju@mit.edu; byildiz@mit.edu

D eep learning based on neural networks has raised tremendous attention as an approach to accelerate machine learning applications, such as computer vision and natural language processing[1,2]. However, using state-of-the-art GPU/CPU (Graphics Processing Unit/Central Processing Unit) based on CMOS (Complementary Metal–Oxide–Semiconductor) circuits to simulate neural networks requires large memory space and high power consumption[3]. This is limited by the traditional von-Neuman structure of current computer systems, which requires the transfer of large amount of data between memory and CPU[4]. New hardware structures and algorithms, such as in-memory computing, have been proposed to tackle these issues[5–8]. As demonstrated by Jo et al.[9], crossbar-type arrays containing two-terminal resistive switches (or memristors) enable energy-efficient and fast-processing physical neural networks[10–12]. The core component of this physical neural network is the resistive switch, whose electronic conductance can be modulated electrically[8,13,14]. This modulation emulates the strengthening and weakening of synapses in the brain. Capturing this synaptic strengthening or weakening, so-called potentiation or depotentiation behavior, respectively[15], is essential to brain-inspired analog computing, and can accelerate complex computation where precision is less critical[2,16,17]. Arrays of resistive switching units have also been demonstrated for image recognition, for example, of faces and hand-written digits[18,19].

There are two main types of resistive switching mechanisms discussed and used in literature to change the conductance, both of them relying on two-terminal configurations[20]. The conductive-filament (CF) mechanism is based on the formation of a narrowly confined conducting path (made of metal atoms or oxygen vacancies) inside an insulating matrix, typically an insulating oxide[21–24]. The phase-change mechanism (PCM) is realized by changing the material between its conducting and insulating phases, such as the Ge-Sb-Te chalcogenides, controlled by Joule heating[25,26]. Both resistive switching mechanisms have been demonstrated in artificial intelligence applications, but the CF mechanism suffers from lack of reproducibility, while the PCM suffers from high energy dissipation and drift[10,27,28]. As a result, the performance of such two-terminal devices in a memory matrix environment is still far from the desired specifications for reliable, fast and energy-efficient training of neural networks[29]. A fundamentally new working mechanism is desirable for enabling new opportunities to address these challenges.

Recently, three-terminal electrochemical resistive switches demonstrated by Fuller et al. exhibit promising characteristics, in particular multistate capability and low energy consumption[30]. This concept relies on a configuration similar to a Li-ion solid-state battery. $LiCoO_2$ was employed as the switching media, whose conductivity can be modulated by the extent of Li intercalation. Gated by a Si layer and separated from the $LiCoO_2$ channel by a $Li^+$ conducting solid electrolyte, $Li^+$ was pumped into and out of the $LiCoO_2$ active material reversibly. Electrochemical control of $Li^+$ concentration in $LiCoO_2$ enables a large and continuous modulation of electronic resistance. Tang et al. also demonstrated high speed and low power modulation of $WO_3$ channel conductance by $Li^+$ intercalation using 5 ns pulses[31]. The major challenge preventing the scale up of this concept is the utilization of Li as the doping ion. Li is not compatible with current CMOS processing due to its high volatility and manufacturing tool contamination concerns.

Being a smaller and faster diffusing cation than $Li^+$, employing protons ($H^+$) as the doping ion presents potential advantages. The Shannon–Prewitt crystal ionic radius of $H^+$ is $-0.04$ Å, in contrast to 0.90 Å of $Li^+$. The advantages of using protons include lower energy consumption, higher operation speed, better compatibility with Si technology, and longer

lifetime due to enhanced structural stability of the active material as a result of volume change with successive ion insertion/extraction steps[30–32]. Electrochemical proton shuffling has been implemented by van de Burgt et al.[32] using organic electrode materials, in a three-terminal electrochemical resistive switch configuration. This device demonstrated greater energy efficiency (390 aJ/$\mu m^2$) and operating speed (order of ms) than its Li counterpart (~37 nJ/$\mu m^2$, order of seconds)[30]. Array-level device demonstrations based on this working mechanism have also been reported, for face recognition and hand-written digit recognition[18,19]. However, the organic nature of the active channel material presents challenges in Si-compatible device fabrication and in long-term stability. While these reports are encouraging, it is desirable to work with a new material system that offers both compatibility with current semiconductor processing protocols and is capable of implementing highly energy efficient analog neural networks. Advancing towards such a technology needs insights from fundamental studies of material atomic and electronic behavior during the resistance modulation process, as well as development of an appropriate gating protocol and device geometry optimization.

In this work, we demonstrate proton intercalation in inorganic materials as a basis for emulating synapse behavior with low energy cost, long retention, and good symmetry. This system offers a deterministic, charged-controlled mechanism that uniformly switches the channel conductance. Thus, it should not pose issues related to lack of reproducibility that heavily depends on the microstructural and chemical heterogeneities in the high-k dielectric materials used in two-terminal CF devices. In addition, if the intercalating ion and the chosen materials are suitable, the energy consumption of the switching process can be very small, as demonstrated in this work. We quantify the protonation-conductance relationship of this artificial electrochemical synapse, providing insights into the proper selection of operating window for the chosen material, $WO_3$. We adopt the constant current gating, and propose that it is more suitable than the constant voltage gating for such three-terminal electrochemical resistive switches because it offers better controllability, reproducibility, and symmetry. X-ray absorption and photoelectron spectroscopy, together with first-principles calculations, reveal that the electronic carrier density and mobility contribute to the conductivity change as a function of protonation in $WO_3$.

## Results
**Material selection and device configuration.** A variety of metal oxides are known to change their electronic conductivity upon cation intercalation, including $WO_3$[17], $MoO_3$[33], $VO_2$[34], and $SrCoO_2$[35]. We chose $WO_3$ as a prototypical proton intercalation host because it is a semiconductor in its undoped state (band gap 2.8~3.2 eV)[36]. We can precisely tune the conductivity by the degree of proton intercalation in this material due to the accompanied filling of electrons into the conduction band[37,38]. With a high concentration of protonation, a tungsten bronze phase ($H_xWO_3$) is expected to form that turns the material metallic[31,39,40]. $WO_3$ is also compatible with Si technology[41]. Protonation of such oxides in the past has been demonstrated from gas-phase $H_2$, through an aqueous electrolyte, by hydrolysis of water adsorbed in a porous solid, or through an ionic liquid electrolyte[17,42,43]. Reliance on protonation from gas-phase hydrogen or from water is not conducive to controllable, reversible, and technologically feasible devices. Instead, we use a solid, inorganic and reversible hydrogen reservoir, palladium hydride, $PdH_x$, that is integrated into a three-terminal device as the gate electrode.

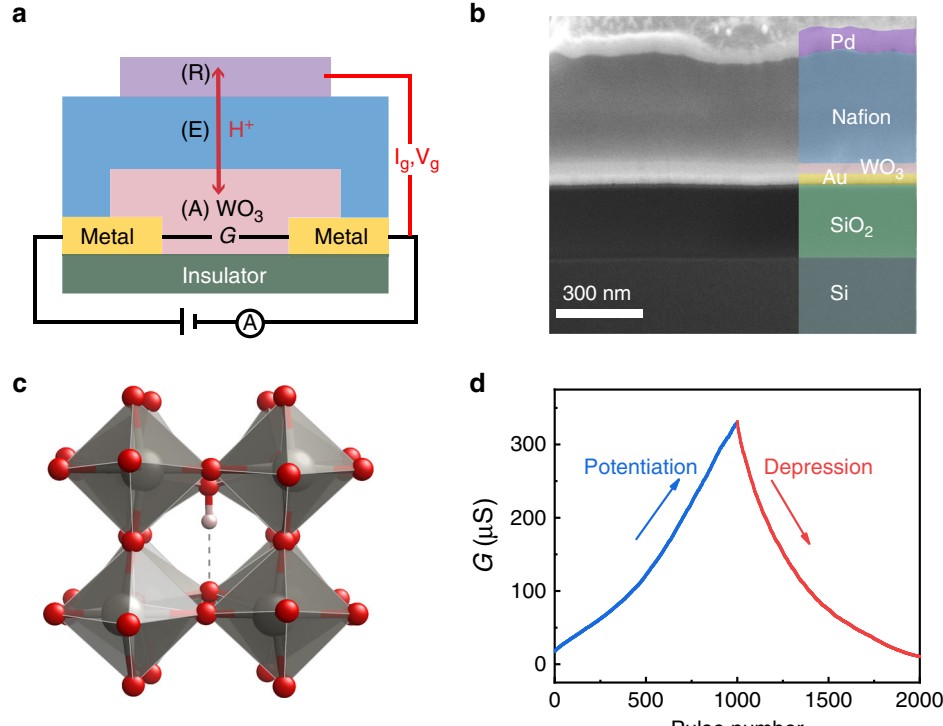

**Fig. 1 Device configuration and operating principles. a** Schematic of the protonic electrochemical synapse device structure. R: solid-state hydrogen reservoir layer, serving also as gate electrode. E: solid-state proton conductive electrolyte layer. A: active switching layer, serving as conducting channel. $G$ refers to the conductance of the channel between the source and drain electrodes (noted as Metal), calculated from $G = I_d/V_{ds}$, where $I_d$ is the current measured at the drain and $V_{ds}$ is the reading bias applied between the source and drain. $I_g$, $V_{gs}$: the gating current and gating voltage, applied between the gate and the source. **b** Scanning electron microscopy image of the device cross-section. **c** $WO_3$ lattice indicating the sites that allow proton intercalation and the bonding between the proton (white sphere) and the oxygen ions (red spheres). The grey spheres are Tungsten atoms. **d** Demonstration of potentiation/depression behavior of the electrochemical synapse by signals of $+200$ nA/$-200$ nA and 5 ms; potentiation and depression represent the synaptic strengthening and weakening behavior. Each data point represents the average conductance value of the steady states measured after each gating pulse.

The three-terminal source-drain-gate configuration of the device in this work is shown in Fig. 1a, b. The source and drain electrodes are two gold metal contacts deposited on $SiO_2$ coated Si gapped by a 100 μm long channel. This channel is filled with a 50 nm thick $WO_3$ film by reactive sputtering as the active material. On top of the channel material, a proton-conductive solid-polymer electrolyte (Nafion-117) is deposited by spin coating, resulting in 300–400 nm thickness. This electrolyte layer is electronically insulating but protonically conductive, with the proton conductivity around 0.09 S/cm at room temperature under 100% relative humidity[44]. The gate electrode in this device serves the dual role of reservoir of protons and electrons. Metals with high electronic conductivity and high hydrogen solubility are desirable for this purpose[45]. Palladium metal, which can absorb hydrogen effectively even at room temperature[46], was selected as gate in this demonstration. After the deposition of Pd, hydrogen was introduced to form $PdH_x$ by exposing the device to 5% forming gas at room temperature prior to and during the test.

**Operation principles**. The conductance between the source and drain electrodes represents the strength of the synapse in a neural network. In this demonstration, a small constant voltage ($V_{ds} = +0.1$ V) is applied between the source and drain electrodes. The corresponding current signal ($I_d$) is recorded to calculate the conductance ($G = I_d/V_{ds}$). To modulate the conductance, positive or negative gate current pulses ($I_g$) are applied to drive protons into or out of the channel material, respectively.

When $I_g > 0$, $PdH_x$ is electrochemically oxidized and releases protons to the Nafion electrolyte. The protons diffuse through the Nafion electrolyte toward the channel oxide, under the driving force of the gating voltage ($V_{gs}$) between the gate and channel. Once the protons reach the surface of $WO_3$, they intercalate into the $WO_3$ with the simultaneous electron injection from the outer circuit (Supplementary Fig. 1). This process is fully reversible and can be expressed with the following electrochemical reaction:

$$\text{Gate}: PdH_{x+x\prime} = PdH_{x\prime} + xH^+(\text{electrolyte}) + xe^-(\text{circuit})$$

$$\text{Channel}: xH^+(\text{electrolyte}) + xe^-(\text{circuit}) + WO_3 = H_xWO_3,$$

where $e^-$(circuit) are electrons that are metered and pumped via the outer circuit. The net effect after this positive gate current $I_g$ biasing is the intercalation of hydrogen (proton + electron) into the $WO_3$ channel material. Electrons and protons must enter/exit the channel material simultaneously to maintain global charge neutrality. When the external circuit is cut off (gate is open circuit), the electronic insulating nature of the electrolyte prevents the back flow of electrons, and consequently that of protons too. This retention of protons and electrons in the channel material is key to realizing the nonvolatile nature of this device.

The proton as a small cation could reside in the interconnected channels of the $WO_3$ lattice and bond with the oxygen ion to form $OH_O$ defect in these channel sites (Fig. 1c). There is a main difference here with respect to the conductive filament (CF) or phase change mechanism (PCM) approach. That is, the intercalation of the protons involves a high degree of spatial

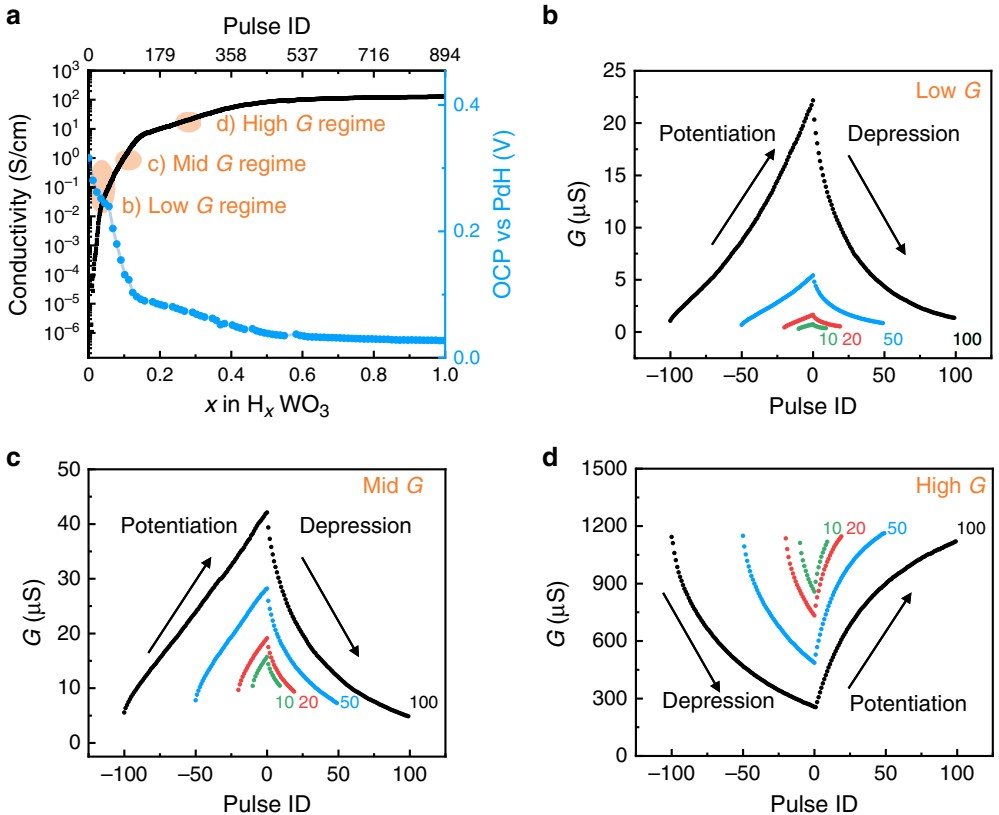

**Fig. 2 Range of tunable conductivity, and electrical potentiation in different conductivity regimes. a** Dependence of electronic conductivity and open circuit potential (channel $WO_3$ vs. gate $PdH_x$) on the hydrogen content in $H_xWO_3$. **b–d** Reversibility of electronic conductance, and symmetry of potentiation (protonation)–depression (deprotonation) behavior, obtained in the three different protonation regimes starting from (**b**) low conductance regime, (**c**) medium conductance regime and (**d**) high conductance regime (depression first to avoid saturation). 10, 20, 50, and 100 states (upon 10, 20, 50, and 100 pulses, respectively) were obtained for each regime by constant current pulsing for potentiation and depression on the same device. Different conductance regimes were achieved by continuously sending constant current pulses with 0.5 μA amplitude and 5 ms width until desired conductance values is reached.

homogeneity within the channel material. This means that the active material ($H_xWO_3$) can stay single-phase during the intercalation due to the high equilibrium solubility limit of H in the solid solution phase.

As discussed above, and unlike the previous protonation of oxides which rely on the electrolysis of water to generate the protons[47], our device is a closed system with proton shuffling between the solid-state gate and the channel material. This not only reduces the energy consumption by avoiding the water hydrolysis reaction, but also improves the controllability. In a practical device, this electrochemical synapse should be encapsulated by insulating layers that are also proton barriers in order to prevent the long-term loss of protons or ingress of oxygen into the system, essential for device endurance. (See Supplementary Figs. 6 and 14 for demonstration of the encapsulated device).

**Electrical response**. The full range of the conductivity dependence on the hydrogen concentration in $WO_3$ ($H_xWO_3$) that is measured in our device is shown in Fig. 2a. A potentiation pulse train was applied to the device with constant current (0.5 μA) and fixed width (5 ms) for each pulse. Between two pulses, the gating circuit was at open circuit for 1 s, during which both the channel conductance ($G$) and the open circuit potential (OCP, between the gate and channel) were measured. With this process, we increased the H content ($x$) from nominally 0 as in $WO_3$ ($W^{6+}$) to the theoretical maximum of 1 as in $HWO_3$ ($W^{5+}$). The H

content ($x$) was estimated from the net charge transferred by the gate, which was evaluated by integrating the gate current over the pulse time and assuming 100% Coulombic efficiency. As demonstrated in Fig. 2a, over seven orders of magnitude change in conductivity was achieved across the full protonation range.

The slope of conductivity vs. $x$ is not a constant across the full range. When the conductivity is plotted in log scale, as done in Fig. 2a, there are clear changes in the slope at $x = 0.03$, 0.15, and 0.5. We ascribe these to the boundaries between three different regimes of protonation, that correspond to a low conductance regime, a medium conductance regime, and a high conductance regime. These are designated as Low $G$, Mid $G$ and High $G$, respectively, on Fig. 2a. The dependence of the conductance slope on the hydrogen concentration implies different conduction mechanisms. This can arise from factors such as defect–defect interactions as the proton concentration increases, or crystal structure change. The slope of the OCP also changes as a function of protonation regime, as seen in Fig. 2a. This implies the involvement of phase change as transitioning from one regime to another (Supplementary Discussion 3 in Supplementary Information).

The electrical response of the device, including the ratio of the maximum to minimum conductance achieved ($G_{max}/G_{min}$) and the symmetry of potentiation–depression, depend on the degree of protonation in $WO_3$, or the corresponding conductivity regime, as shown in more detail in Fig. 2b–d. The low hydrogen content (low conductance, Fig. 2b) regime offers better symmetry and higher $G_{max}/G_{min}$ ratio. The high hydrogen content (high conductance) regime features larger changes in the absolute

conductivity per pulse (per the same amount of change in the H quantity) at the cost of poor symmetry. These results guide the choice of an operating window for the device that depends on the performance matrix that is most desirable.

To demonstrate reversibility, constant current pulse trains containing different polarity but the same amplitude (0.5 μA) and width (5 ms) were applied between the gate and the channel. When the protonation of $WO_3$ is confined to the low conductance regime shown in Fig. 2b, after 100 positive gating pulses, the conductance of the channel increased from 1 μS to 22 μS, demonstrating a continuum of 100 states and a $G_{max}/G_{min}$ ratio of more than 22 (black trace). Upon the application of an identical but negative current pulse train, the conductance decreased to the original low state, accomplishing the depression process. The number of states achieved during this process can be tuned with different number of pulses, and with the pulse height–width that control the degree of protonation in each operation. The states need to be distinguishable from each other by the electronic resistance measurement apparatus. Based on the current device and experimental set up, we have demonstrated 1000 states, with an average $\Delta G/G$ of 0.3% per pulse and an overall $G_{max}/G_{min}$ of 30, as shown in Fig. 1d. These values are in line with the desirable specifications of resistive processing units as assessed by Gokmen et al.[29], and can be further improved with optimization of device geometry and electronic measurement sensitivity. The $G_{max}/G_{min}$ ratio also depends on the pulse number, being 2, 4, 8, and 22, for the 10 (Green), 20 (Red), 50 (Blue) and 100 (Black) pulses, respectively, in the low-$G$ regime as shown in Fig. 2b.

An identical procedure in the medium and high conductance regimes (higher H content) gives smaller $G_{max}/G_{min}$ ratios, as shown in Fig. 2c, d, respectively. Taking the switching behavior of 100 pulses as an example, the three regimes denoted as Low $G$, Mid $G$, and High $G$ result in distinctly different $G_{max}/G_{min}$ ratio of 22, 8, and 4, respectively. In addition, a less symmetrical shape was observed in the higher conductance regime. Comparing the 10, 20, 50, and 100 pulse behavior in either protonation regime, a shallower protonation/deprotonation in $WO_3$ allows better reversibility (the potentiation/depression symmetry). But this comes at the expense of a smaller conductance $G_{max}/G_{min}$ ratio. When compared with the symmetry behavior in prior demonstrations of the two-terminal conductive filament resistive switches[9,48], a significant improvement of symmetry is observed here.

There is still a small extent of hysteresis in our potentiation–depression data. This originates from the long time scale of the equilibration of the protonated state due to proton diffusion from the Nafion-$WO_3$ interface to the bulk of $WO_3$ layer (and vice versa). With the same sampling period applied to all measurements, we had to acquire the non-equilibrated states at higher H content regimes which took longer time to equilibrate. Reducing the channel layer thickness (smaller diffusion length) and reducing the pulse width and height (smaller amount of H exchange per pulse) will help to accelerate the kinetics of relaxation and improve the symmetry further. As seen in Fig. 1d, when the pulse amplitude is 200 nA, the inserted hydrogen amount during each pulse is less, and the equilibration is faster, thus giving a more symmetric potentiation/depression behavior even with 1000 pulses, compared with the behavior in Fig. 2 obtained by 0.5 μA pulse amplitude.

The average voltage resulting from the gating pulses is shown in Fig. 3a. With the pulse current of ±0.5 μA, the resulting gate voltage in each potentiation step, $V_{gs}$ is smaller than ±0.5 V. The energy consumption per unit change in conductance, $\Delta G$, during the gating process can be estimated by the integration of the charge vs. voltage. With the average gating voltage of 0.25 V resulting from current pulses of 0.5 μA and 5 ms, an average $\Delta G$

of 48 nS, and an area of $0.6 \times 1.2 \ mm^2$, the average energy cost per unit area per unit conductance change is only 18 aJ/(μm$^2$ × nS). This value is significantly smaller than the energy consumption of the conductive filament or phase change mechanisms, and similar to that of state-of-the-art protonic organic devices recorded in the literature[32].

The low operating voltage of <0.5 V also proves that wo do not rely on the water electrolysis to supply the protons. To electrolyze water, a minimum of 1.23 V is necessary to meet the thermodynamic requirement and an additional 0.5–0.7 V is needed to break the kinetic overpotential[49], even on the best electrocatalysts. So, we can confidently rule out the possibility of water electrolysis here.

Two features of this protonic electrochemical synapse discussed here enable this low operating energy. First, the cation employed here to modulate the conductivity is the smallest and lightest cation, the proton, needing low energy to migrate and transfer at interfaces. Second, the electrochemical potentials of the gate and channel materials are close, resulting in a small open-circuit potential, as shown in Fig. 2a. This means that the device is operated only within a small voltage window. The first feature minimizes the kinetic contribution to the energy consumption, while the second one minimizes the thermodynamic contribution. In addition, the small but non-zero open-circuit potential change after gating indicates that a selector component may be needed to program individual synapses in an array, as demonstrated in ref. [19], and to avoid cross-talk of individual synapses when not gating.

To demonstrate the endurance of the device, a repeated potentiation and depression cycling test was performed and shown in Fig. 3b. Each cycle contains 100 potentiation and 100 depression pulses. This process was repeated 100 times. The $G_{max}/G_{min}$ ratio remained consistent around 3 across the 100 cycles, demonstrating a device endurance >20,000 pulses. Additional endurance tests at different conductivity regimes, number of states, and encapsulation of the device are provided in Supplementary Figs. 3–6. Potential failure mode of this type of device are the ingression of oxygen, out diffusion of hydrogen, <100% Faradic efficiency, and electrode degradation. Each of these modes of failure can be overcome by proper encapsulation, that warrants future work. (See Supplementary Discussion 2).

**Resistance modulation mechanism.** To probe how protonation changes the electronic structure of $WO_3$, we investigated its valence band structure by X-ray photoelectron spectroscopy (XPS). Before proton intercalation, the valence band of pristine $WO_3$ features a broad peak located around 7 eV below the Fermi level (Fig. 4a)[37]. The extrapolation of the upper edge of this band leads to a valence band maximum located ~2.5 eV below the Fermi level (binding energy = 0 eV) and the photoelectron emission is nearly zero at the Fermi level. This measurement confirms the semi-conductive nature of unprotonated $WO_3$, with the Fermi level positioned within the band gap. After proton intercalation (~$H_{0.08}WO_3$, see details in Supplementary Figs 7–8 for sample configuration used in X-ray spectroscopy experiments), a new prominent peak appears at nearly 0.5 eV below the Fermi level, indicating the filling of electrons in W $5d(t_{2g})$. The photoelectron emission is also non-zero at the Fermi level. These two observations indicate the filling of electrons into the previously empty conduction band and transition to a metallic conduction mechanism[4].

The oxidation state of W was probed by the W $4f$ photoelectron peaks in Fig. 4b. For the unprotonated $WO_3$, the binding energy of the of W $4f_{7/2}$ and W $4f_{5/2}$ peaks are located at 35.9 eV and 38.0 eV, respectively. This spectrum is consistent

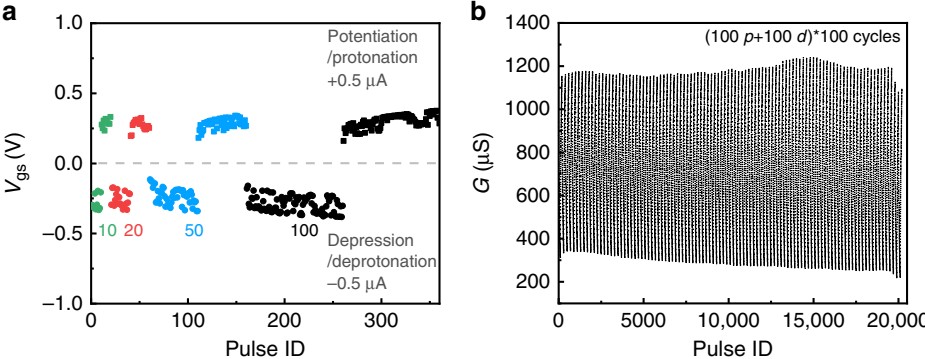

**Fig. 3 Gate voltage restulting from gate current pulses, and retention of multistate capability upon cycling. a** Average potentiation/depression voltage, resulting from pulses of ±0.5 μA and 5 ms, during the operation of the device in the medium conductance regime shown in Fig. 2c. **b** The retention of multi state capability over 100 cycles of the device, each cycle contains 100 potentiation and 100 depression steps. These data are obtained in the high conductivity regime.

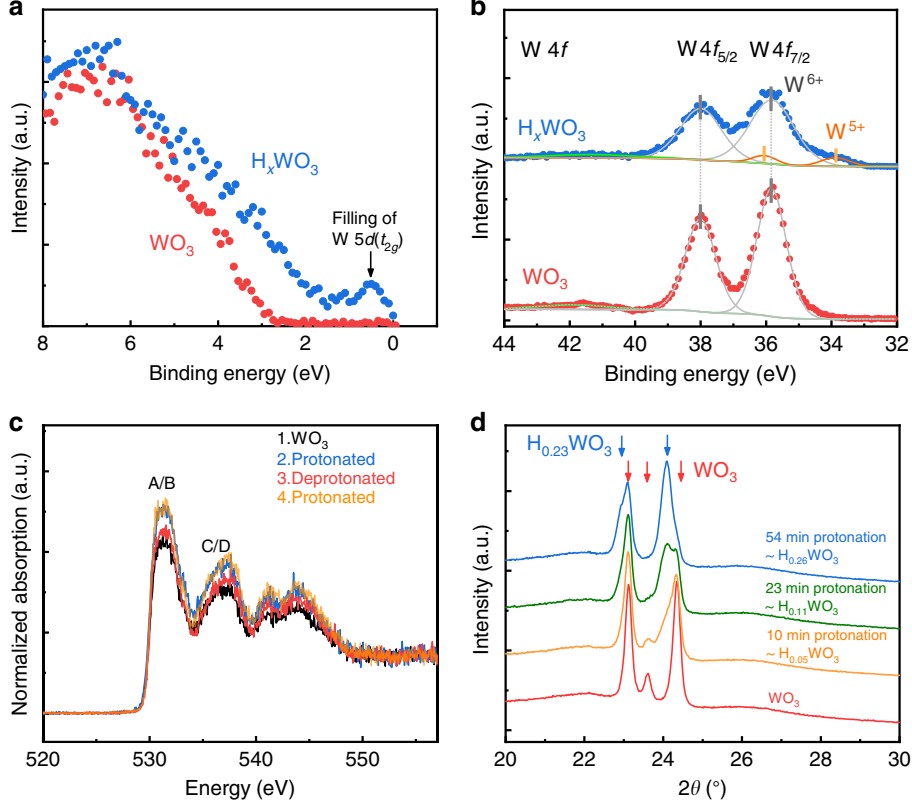

**Fig. 4 Electronic and crystallographic structure during protonation to tune the conductance. a** Intensity of photoelectron emission vs. binding energy, at the low binding energy region corresponding to the valence band of $WO_3$ and protonated $H_xWO_3$, measured by X-ray photoelectron spectroscopy (XPS). **b** W 4f photoelectron emission peak, measured by XPS. The oxidation states of $WO_3$ and protonated $H_xWO_3$ were deduced from the position of the W $4f_{7/2}$ peak. **c** X-ray absorption spectra of the oxygen K-edge, upon two cycles of protonation/deprotonation: (1) $WO_3$, (2) first cycle protonated $H_xWO_3$, (3) deprotonated $WO_3$, and (4) second cycle protonated $H_xWO_3$. **d** Crystal structure change probed by X-ray diffraction pattern with heavy proton intercalation corresponding to the different conduction regimes shown in Fig. 2a.

with $W^{6+}$ [50]. After protonation, a new W $4f_{7/2}$ component at 33.8 eV was identified. This lower binding energy corresponds to the reduced W oxidation state of 5+ [51]. Deconvolution of the peaks reveals the ratio between $W^{5+}$ and $W^{6+}$ to be 1:11.4, corresponding to $H_{0.08}WO_3$.

From the analysis of the valence band and the W 4f photoelectron peak, we can conclude that, upon intercalation of protons, new in-gap states form, gradually lowering the effective bandgap of $WO_3$ to 0. This process is accompanied by the injection of electrons, which pushes the Fermi level toward the

conduction band. When the quantity of injected electrons is significant enough, the Fermi level completely shifts into the conduction band. Consistent with electron injection and n-type doping, the oxidation state of W is reduced. As a result, we expect a continuous increase of electronic conductivity with increasing protonation level in $H_xWO_3$, as shown in Fig. 2a.

In addition to the increase in carrier density, the protonation of $WO_3$ could result in an increase in the mobility of the electronic charge carriers, that may arise from the increase in hybridization of W–O bonds. In operando X-ray absorption spectroscopy of the

O K-edge region measured during the protonation of WO₃ provides strong evidence for this effect. The O K-edge spectra of WO₃ prior to and after protonation is shown in Fig. 4c (see sample configuration in Supplementary Fig. 7). A prominent absorption peak denoted as A/B at the photon energy of c.a. 530 eV corresponds to electron excitation from core level O 1s to the empty states in O 2p, which is created by W 5d($t_{2g}$)-O 2p hybridization[52,53]. The intensity of this peak is positively correlated with the hybridization level (covalency) of the W–O bonds[54]. After protonation (Trace 2, blue), the A/B peak intensity increases compared with the state prior to protonation (Trace 1, black). This indicates a higher level of hybridization after protonation. Because this hybridization is featured by electron donation from the O 2p orbital in the valence band to the W 5d ($t_{2g}$) orbital in the conduction band, the increased hybridization results in a larger extent of electron delocalization[55], which is usually accompanied by higher electron mobility[56]. In addition, the peaks at c.a. 535–537 eV (C/D) reflect the hybridization of the W 5d($e_g$)-O 2p states, and these peaks are sensitive to the O 2p-H 1s interaction[52]. The increase of intensity in C/D peaks after protonation is a direct evidence of proton intercalation into WO₃. The observed changes in the O K-edge are reversible upon cycling between protonation and deprotonation. The peak intensities of the A/B and C/D regions decrease with deprotonation (Trace 3, red), and increase again with a subsequent protonation (Trace 4, orange) (Supplementary Fig. 8 for details). This modulation of the degree of W 5d($t_{2g}$)-O 2p hybridization changes the extent of electron delocalization, and thus, the electron mobility.

First-principles calculations of electronic structure also give evidence to the increase of charge carrier density and mobility upon increased protonation in WO₃[38]. The evolution of the electronic structure in monoclinic $H_xWO_3$ upon hydrogenation ($x = 0$–0.125) is shown in Fig. 5. The calculated direct band gap of pure monoclinic WO₃ is 2.82 eV, in good agreement with the available experimental data[36] and Supplementary Fig. 9. As it is seen from Fig. 5b, the conduction band of monoclinic WO₃ is mainly composed of unoccupied W 5d orbitals, while O 2p orbitals dominate in the valence band region.

Insertion of protons at the channel sites as shown in Fig. 1c, followed by subsequent charge balancing leads to the formation of in-gap states (Fig. 5d), at 0.28 eV below the conduction band for $x = 0.0156$. At low hydrogen concentrations, the excess electrons are localized over several adjacent W atoms located in the same plane forming a large 2D polaron with a radius of ≥15 Å, as shown in Fig. 5c.

With increasing protonation, the localized in-gap states monotonically approach the conduction band minima, decreasing the ionization energy from 0.28 eV at $x = 0.0156$ to 0.08 eV at $x = 0.0625$. Further protonation up to $x = 0.125$ leads to a metallic behavior of $H_{0.125}WO_3$, as shown in Fig. 5e, f. The computed changes in electronic structure correlate well with the decreased oxidation state of W and the disappearance of the band gap at $x = 0.08$ probed by XPS shown in Fig. 4a–c.

At higher protonation levels ($x > 0.1$), the distortion of the lattice may actually change the crystal structure[57], and thus, its band structure. To study this, we performed in operando X-ray diffraction to study the structure evolution. Prior to protonation, WO₃ exhibits the most stable monoclinic structure[57], as shown by the red pattern denoted as WO₃ in Fig. 4d, with three major diffraction peaks at 23.1°, 23.6°, and 24.3°. They correspond to the (002), (020), and (200) planes (JCPDS 83-0950). During continuous biasing with $I_g = +1$ μA, the (002) and (200) peaks of WO₃ shifted toward lower angles after 10 min ($x = 0.05$). After 54 min of biasing ($x = 0.26$ by assuming 100% Coulombic efficiency, see Supplementary Table 1), these two peaks eventually shift to 22.9° and 24.1°, corresponding to +1.29% and 0.95% lattice expansion, respectively. These two peaks can be related to the (001) and (110) planes of the tetragonal $H_{0.23}WO_3$ (JCPDS 85-0967, Supplementary Fig. 8 for higher angle XRD data). The original (020) peak of monoclinic WO₃ at 23.6° disappeared during the process because of increased symmetry. The total unit cell volume expanded only a very small amount, +0.28% for

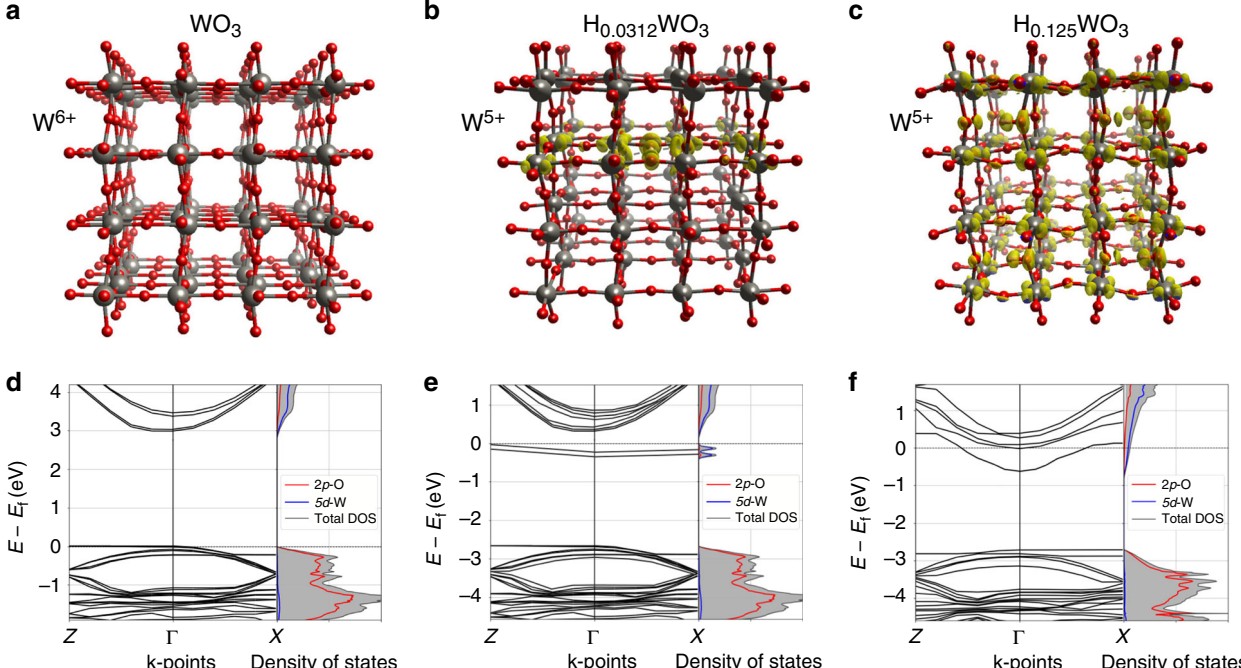

**Fig. 5 Calculated electronic structure with protonation in WO₃.** Crystal structures, charge density maps and corresponding electronic band structure and density of states for (**a**, **b**) WO₃, (**c**, **d**) $H_{0.0312}WO_3$, (**e**, **f**) $H_{0.125}WO_3$; grey atoms—W, red—O and white—H. The excess charge associated with the H insertion is shown in yellow (isosurface – 0.04 e/Å³).

0.23 H per unit $WO_3$. For comparison, the c-axis change from $Li_0CoO_2$ to $Li_{0.25}CoO_2$ is nearly 9.3%[58]. The reduction of lattice stress further highlights the advantage of $H^+$ over $Li^+$ in terms of structural stability[59].

As a result of this in situ XRD analysis, we can conclude that, with the insertion of protons into the channel sites, the $WO_6$ octahedra become more ordered and the crystal structure becomes more symmetrical, as seen from the monoclinic to tetragonal transition in Fig. 4d. The increased symmetry can arise from increased concentration of electronic charge in $WO_3$, namely $W^{5+}/W^{6+}$ polarons. We have also found that the interaxial angles approach 90° upon proton insertion in $WO_3$ (see Supplementary Fig. 11 in Supplementary Information.) The increased symmetry of the structure can also contribute to a change of the band structure[60] and eventually the conductive behavior[39,57].

**Gating strategy of electrochemical synapses**. Gating through the application of a voltage is widely adopted in the literature to induce resistive switching (also demonstrated in Supplementary Fig. 12 on our device)[61]. Typical reported gating voltage values range from several volts to tens of volts. Very often, the gating voltage is used as a descriptor to indicate the final state of the sample[49,62]. Here we emphasize that, in such an electrochemical synapse (or ionic gating) devices, there can be a significant difference between the gating voltage applied and the change in the open-circuit potential of the device due to polarization. That is, the equilibrium chemical potential of the active material does not necessarily follow the applied gating voltage precisely[63]. Such difference could be a major reason for the lack of consistency of the gating voltages reported in the literature[17,40,49,64].

There are two major points to consider. First, the voltage applied during the ionic gating is spent on several contributions; the chemical potential difference between the gate and the channel electrodes, ionic ohmic loss across the electrolyte, and the Nernst overpotential to drive the electron-ion exchange at the electrode–electrolyte interface. The steady-state chemical potential of the active material achieved after the constant voltage gating represents only the first part, and the latter two are the kinetic loss terms.

Second, even if the correlation between the gating voltage and the chemical potential at the electrode–electrolyte interface can be properly established, the time needed to reach equilibrium within the bulk of the channel material is non-negligible. This is due to the finite diffusion time of ions in solids (Supplementary Fig. 13).

Based on these considerations, we propose two directions to evade the issue related to voltage gating. First, as adopted in this paper, constant current gating, rather than constant voltage gating, should provide a deterministic switching behavior, by using the accumulated charge as the descriptor for the status of the channel. Second, employing cations with high diffusivity, such as the proton, and reducing the thickness of the channel layer, should minimize the diffusion time and the resulting time-dependent relaxation of the channel conductance.

In summary, we have designed and demonstrated a protonic electrochemical synapse as an all-solid-state device, with very low energy consumption $(18 \text{ aJ}/(\mu m^2 \times nS))$, nearly symmetrical potentiation (protonation)/depotentiation (deprotonation) behavior, long cycling lifetime, and featuring small changes in OCP. The core material system comprising $WO_3$ as the channel and $PdH_x$ as the gate electrode (solid proton reservoir), were chosen in order to be CMOS compatible. In the complete range from 0 to 1 H per $WO_3$, we achieved continuous modulation of the conductance and a high $G_{max}/G_{min}$ ratio of $10^7$. The lower protonation regime features higher potentiation/depotentiation

symmetry. The higher protonation regime features small OCP change and large $\Delta G$ per unit amount of proton insertion. This understanding provides guidance for selection of an operating window to achieve desired device properties. In addition, we confirmed the topotactic nature of the conductivity modulation of our device through electronic and atomic structure analysis, with the observed volume change much smaller than Li-intercalation compounds, thus promising smaller residual stress and longer cycle life. The inorganic protonic electrochemical synapse demonstrated here, and the understanding generated from this study, provide a path to low-energy and reliable artificial synapses as hardware for artificial neural networks that are capable of accelerating artificial intelligence development.

## Methods

**Device fabrication**. The device adopted a three-terminal thin-film transistor configuration. The source and drain electrodes composed of 40 nm thick Au was deposited by RF sputtering. A channel with a width of 500 μm and length of 100 μm between the source and drain electrodes was patterned by shadow mask. This channel was further filled by 50 nm thick $WO_3$, which was deposited by reactive RF sputtering from W metal target with the $Ar:O_2$ ratio of 9.3:2.7 at 3 mtorr. The deposited film was further annealed at 450 °C for 1 h in air to achieve crystalized monoclinic $WO_3$ phase. Spin coating was employed to apply Nafion-117 resin solution (Sigma-Aldrich) on top of $WO_3$ as solid proton conductor layer. The thickness of the proton electrolyte layer can be adjusted by the spin speed and concentration to achieve 300–400 nm in thickness. After drying the Nafion layer in ambient air followed by 100 °C baking, the device was transferred back to the sputtering chamber for Pd top gate electrode deposition patterned by shadow mask.

**Electrical measurements**. The as-fabricated device was transferred into a moisturized chamber filled with 5% hydrogen in argon atmosphere and nearly 100% relative humidity at room temperature. The hydrogen absorption by Pd can be observed by continuous open circuit measurement of $WO_3$ (channel) vs. Pd (Gate), which changes from negative to positive, due to the electrochemical potential change when Pd is converted into $PdH_x$. The electrical test was performed inside the chamber without exposing to the ambient air. A constant voltage of +100 mV was applied between the source and drain, the corresponding current ($I_d$) was recorded by Keithley 2460 to calculate the conductance level. A constant current pulse train with ±0.5 μA amplitude, 5 ms width and 1 s interval were applied by Gamry reference 3000 potentiostat or Keithley 2400 source meter to the gate to induce the potentiation and depression characteristic behavior of resistive switching. The gate voltage was recorded every 1 ms during the pulse to calculate the energy consumption per pulse. Between pulses, the gate electrode was set at open circuit condition (high impedance mode) for OCP measurement for 1 s.

**Material characterization**. X-ray diffraction pattern was collected on Rigaku Smartlab diffractometer (Cu Kα λ = 1.5406 Å) in situ. A device with the same configuration as described above but with larger size ($WO_3$ pad: 8.3 mm × 5 mm × 100 nm) was prepared for this test to ensure the intensity of diffraction signal to be strong enough. The device was encapsulated in a sample holder with X-ray transparent windows made by Mylar film. The chamber was filled with 5% $H_2$ in Argon and the electric bias was applied by Keithley 2460.

The ambient pressure soft XAS data was collected at the 23-ID-2 (IOS) beamline at the National Synchrotron Light Source II (NSLS-II) at Brookhaven National Laboratory. Partial fluorescence yield signal, measured using a Vortex EM silicon drift detector, was chosen as the signal to be collected due to its relatively large probing depth (~100 nm). With this probing depth, we consider the data collected to be sensitive to the bulk of film, given that our film is typically 50 nm thick. Detection area was carefully chosen at the edge of the $WO_3$-Au interface (See Supplementary Fig. 7). The device was first transferred into the in operando ambient pressure XAS chamber, after which 5 mTorr $H_2O$ and 400 mTorr $H_2$ was introduced into the chamber. A reference spectrum was collected without external bias, and denoted as $WO_3$ in Fig. 4c. A continuous negative current was further applied on the $WO_3$ through the Au contact. It is expected that hydrogen will split into hydrogen atom on the Pd side, which can be electrochemically pumped into $WO_3$ through the electrolyte, even under such low hydrogen partial pressure.

The XPS measurement and valence band measurement were collected by PHI Versaprobe II spectrometer with Al Kα source. XPS measurement was taken ex situ for the same sample after XAS measurement at the protonated state.

The cross-sectional samples were prepared by FEI Helios Nanolab 600 focused ion beam system and imaged in the same instrument by electron beam.

**Density functional theory calculations**. First-principles calculations were performed within the density functional theory formalism using the projector augmented wave potentials[65] as implemented in the Vienna Ab initio Simulation

Package [66]. The screened Heyd-Scuseria-Ernzerhof hybrid functional for solids[67] was employed to accurately describe many-electron interactions, charge localization, and lattice parameters The calculation were performed for monoclinic $WO_3$ (space group P21/n) supercell of 14.70 Å × 15.01 Å × 15.41 Å ($\beta = 90.17°$) with multiple H atoms to simulate the concentration range of $x = 0–0.125$. A mesh of $4 × 4 × 4$ **k**-points and a 500 eV cut-off energy were employed for supercell calculations.

## Data availability

The data that support the findings of this study are available from the corresponding authors upon request.

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

## Acknowledgements

This work was supported by the MIT Skoltech Program, by the MIT Quest for Intelligence, and by the MRSEC Program of the National Science Foundation under award number DMR - 1419807. The device fabrication was carried out through the use of MIT's Microsystems Technology Laboratories and MIT.nano. This work made use of the Shared Experimental Facilities supported in part by the MRSEC Program of the National Science Foundation under award number DMR - 1419807. This research used resources of the 23-ID-2 (IOS) beamline of the National Synchrotron Light Source II, a U.S. Department of Energy (DOE) Office of Science User Facility operated for the DOE Office of Science by Brookhaven National Laboratory under Contract No. DE-SC0012704. We acknowledge the Extreme Science and Engineering Discovery Environment (XSEDE) computational resources provided through allocation TG-DMR190038. We thank Dr. John Rozen, IBM, for technical mentorship and support on this research. We thank Jiayue Wang and Younggyu Kim for help with the XAS experiments, and Yanhao Dong for the discussion on gating approach.

## Author contributions

B.Y., J.L. and J.D.A. designed, initiated and supervised the project. X.Y. designed, fabricated, and tested the device, with help from W.L., M.O., S.R., and N.E. Material characterization was performed by X.Y., with help from D.K. on the XPS. I.W. and A.H. contributed to the XAS measurement and data interpretation. K.K. conducted the first-principles calculations. All authors discussed the results. X.Y., B.Y., J.L., and J.D.A. wrote the paper.

## Competing interests

The authors declare no competing interests.
