## [Peer Review File · Nature Communications]

Reviewers' comments:

Reviewer #1 (Remarks to the Author):

Emond et al: This interesting paper deals with a WO₃-based device, the conductivity of which can be modified by electrochemically pumping hydrogen in and out. I.e., the conductivity of the channel can persistently be changed back and forth, not only on/off but with multiple steps in between. Such devices termed "electrochemical synapses" could be very beneficial for analog implementations of neuronal networks. The principle has been realized before with other materials (organic semiconductors, or WO₃ with lithium insertion) but the present combination of WO₃ with hydrogen pumping is expected to be advantageous with respect to long-term stability and energy consumption.

The physical mechanism of changing WO₃ conductivity by hydrogen insertion=reduction is straightforward (and well supported by XPS and DFT results), also the appearance of the different regimes in fig. 2a.

It is important to explore several options how to realize such devices (which channel material, modified by which ion insertion, which operation mode...), and thus overall I recommend publication.

Some points related to the practical applicability should be discussed in more detail:

- * regarding the multi-state capability, authors should be more specific which magnitude of conductance change is regarded necessary for reliable and reproducible access of one state, and how many states can realistically be implemented in one cell
- * low hysteresis is decisive for multi-state operation. I'm a bit surprised that even under current control quite big hysteresis occurs under some conditions, e.g. fig. 2c,d. Any idea why, and how to improve that?
- * According to the reported results, current gating is preferred as operation mode over voltage gating. Please comment if this would lead to any significant complications for construction of real devices comprising many such synapse cells (current is more difficult to control than voltage), or how big the chance is to get good performance also in voltage gating
- * it would be good to comment on the scatter in conductance between a larger number of nominally identical cells (i.e. how reliably can the number of cells be scaled up)
- * the cells need humidity in the Nafion layer, and oxygen in- and/or hydrogen out-diffusion must be avoided. Authors report some preliminary encapsulation tests, but longer times than in fig S14 should be reported
- * a related issue: Nafion is an acidic polymer. Any long-term degradation in contact with palladium hydride (the hydrogen reservoir) by releasing hydrogen from proton+hydride ions, and incorporation of Pd²⁺ into Nafion for charge compensation?

Reviewer #2 (Remarks to the Author):

In this manuscript the authors present a very complete work on the development of three-terminal synapses based on proton intercalation in inorganic materials. The work is very complete, as it provides clear information about the fabrication of the devices, complete morphological and electrical characterization, and atomistic modelling. The electrical characteristics measured are impressive, specially the linearity achieved and the controllability of the potentiation and depression processes. The conductance modulation mechanism is clearly identified using "in operando" chemical measurements. Overall, the work provides important insights in its field of research, and the reliability of the findings is higher than in most reports in this field due to the multiple and high quality techniques used. In my opinion, this work is above the level of most others in the field, and for that reason I would like to recommend its publications in Nature Communications.

The only weakness that I see, which the authors should definitely further discuss, is the integration capability of this work:

1- First of all, they use devices with channels of 100 μm (is that the length or the width?). These devices are very big. Did the authors try to do smaller devices? This is highly recommended in order to observe if the findings also apply to miniaturized devices.

2- Moreover, as the authors wrote the word "computing" in the title of their abstract and because the introduction of their paper is discussing about "in memory computing", the authors should discuss what is the potential of their device for computation, which in its current form seems to be unclear to me. The authors should clarify which kind of computation may be done with their device and the path towards such implementation. I do not think it is necessary they demonstrate such computation, but they may reference some works that did such things.

3 - The authors should also indicate which is the advantage and disadvantage of their devices compared to two-terminal resistive switching devices. I think two-terminal resistive switching devices are the most commonly used in most companies, and therefore it seems to me that they have most potential to end up being the hardware for such "in memory computing". Therefore, from a technological point of view, the authors should clarify the potential of their work.

The work is really good, I really enjoyed reading it, but the authors should provide more practical information about the technological potential of their work. I also recommend to remove the word "computing" from the title; some readers may get too excited when seeing the title and later disappointed when reading the manuscript, as there is not computing in this manuscript. In the future, the authors may try to associate different devices to solve a complex problem. They may even indicate such thing in the conclusions to encourage the community to work in such application.

Reviewer #3 (Remarks to the Author):

Authors in this work demonstrate a three terminal device based on proton migration. The device shows a great reversibility for analog signal processing under current pulses. They have also performed impressive and careful materials studies both experimentally and theoretically to conclusively reveal the switching mechanism. It is a very interesting and timely work. I strongly recommend it for publication.

I do have some minor comments for the authors to consider.

1. The authors obtained good XAS and XPS results to make comparisons between WO_3 and HxWO_3 , but the exact ratio of O to W of the as-deposited film was not revealed. It would be great if the stoichiometric ratio of O to W is also given in the manuscript. Did author fabricate the WO_x film with a non-stoichiometric ratio? How would that affect the performance of the device? How to eliminate the effect of oxygen vacancies migration in WO_x channel?

2. The WO_3 film was annealed in air at 450 $^\circ\text{C}$ for 1 h. Please comment on the concentration and effects of nitrogen in the annealed film. In addition, 450C is higher than 400C, which means this process is not 100% fab compatible yet. Would 400C annealing for a longer time achieve the same effect?

3. Please comment on how to control concentration of hydrogen in Pd, nafion and WO_3 during fabrication process? How do we know that H has been incorporated into the Pd films? Can phase/structure of PdH_x or Hydrogen forward scattering (HFS) help?

4. The relative humidity was 100% during the testing. Does it mean that the devices still somehow rely on water electrolysis? Would any change of the environment humidity affect the proton conductivity of the device? Would the device work well when the test chamber contains hydrogen only? What is the chamber pressure when the devices were tested?

5. In Figure 2b-d, the conductance updating property of the device at low conductance regime

shows a better symmetry than that at middle and high conductance regime. Why does the symmetry of the conductance updating become worse when the concentration of hydrogen increases? Why there is still an asymmetrical updating in low conductance regime? Since the symmetry of conductance updating with 200 pulses looks better than that with 40 pulses and 100 pulses in low conductance regime, would the symmetry be affected by pulse number?

6. In Figure 3b, is there a plausible explanation for that the high resistive state of the device becomes more insulative and low resistive state of the device becomes more conductive after large amounts of cycling?

7. on Page 7, there is a typo, "As demonstrated in Figure 1a, over seven orders ..." should be Figure 2a.

8. on Page 9 about Figure 2a "...after 100 positive gating pulses, the conductance of the channel increased from 1 μ S to 22 μ S, ..." it's a bit confusing here, what is the pulsing sequence, from 100 down counting to 1 in Figure 2b? please use arrows to label them in the figure.

9. The specific pulse height-width that was used to achieve each conductance regime (low, medium and high) should be clearly stated in the caption of Figure 2 or the main text.

10. Endurance to the level of million has not been demonstrated (and not really expected for this paper as well). Is that due to testing time or failure? Please comment on possible failure mode?

11. As the author mentioned, "When the external circuit is cut off (gate is open circuit), the electronic insulating nature of the electrolyte prevents the back flow of electrons, and consequently that of protons too." Therefore, a two-terminal threshold volatile switch was used to provide both selector and threshold functions for programming (e.g. NatMat 16, 396, 2017), without which an array of such devices may only be programmed column-wise or row-wise (not randomly and individually programming a device in the array during learning). Will the device studied here also need such as threshold switch on the gate?

12. A constant current source (compared to a voltage source) is usually not preferred in real circuits due to various reasons.

We thank the reviewers for their constructive comments and support of our work. We have addressed their comments point-by-point in the bullets below, and have made the relevant changes in the manuscript where the changes are highlighted by yellow. We hope our paper is ready for publication in this revised form.

Reviewer #1 (Remarks to the Author):

This interesting paper deals with a WO_3 -based device, the conductivity of which can be modified by electrochemically pumping hydrogen in and out. I.e., the conductivity of the channel can persistently be changed back and forth, not only on/off but with multiple steps in between. Such devices termed "electrochemical synapses" could be very beneficial for analog implementations of neuronal networks. The principle has been realized before with other materials (organic semiconductors, or WO_3 with lithium insertion) but the present combination of WO_3 with hydrogen pumping is expected to be advantageous with respect to long-term stability and energy consumption.

The physical mechanism of changing WO_3 conductivity by hydrogen insertion=reduction is straightforward (and well supported by XPS and DFT results), also the appearance of the different regimes in fig. 2a.

It is important to explore several options how to realize such devices (which channel material, modified by which ion insertion, which operation mode...), and thus overall I recommend publication.

Comment: *Some points related to the practical applicability should be discussed in more detail:*

** regarding the multi-state capability, authors should be more specific which magnitude of conductance change is regarded necessary for reliable and reproducible access of one state, and how many states can realistically be implemented in one cell*

Response: The number of states achievable, the maximum/minimum conductance ratio (G_{\max}/G_{\min}), and the relative change of resistance at each pulse depends on the initial hydrogenation state (including zero) and the pulse parameters. In principle, within the same G_{\max}/G_{\min} , the number of states can be controlled by the amount of charge injection during each pulse. ($G_{\max}/G_{\min} = \text{Number of states} \times \text{Difference between each state, } \Delta G$, in the linear regime). In the literature, the desired specifications for reliable training of neural networks were outlined by Gokmen and Vlassov, *Front. Neurosci.* **10**, 333 (2016). The specs proposed included a change of resistance at each step to be at least $\Delta R=100 \text{ K}\Omega$, and the min value of the low resistance state to be $R=14\text{M}\Omega$, and a minimum of 8-fold tuning of resistance up to $112\text{M}\Omega$. This corresponds to $\Delta G/G$ to be at least $0.7\sim 0.09\%$ $\Delta G/G$ and implies nearly 1000 states should be needed.

Based on our current device characteristics, when the pulsing current at each pulse is sufficiently small, 1000 states can be achieved, as demonstrated in the new Figure 1d. In this specific demonstration, 200 nA/5 ms pulsing was introduced to get an average of 0.3% for $\Delta G/G$ per pulse, generating an overall G_{\max}/G_{\min} of 30. In the meantime, the conductivity detection limit of the instrument is $\sim 0.025\%$ (from specification of Keithley 2460). Therefore, we are confident 1000 states can be achieved on our device. The maximum and minimum resistance values will depend also on the geometry of the device.

We have revised Figure 1d on Page 6 and corresponding main text on Page 10 to reflect this discussion.

Comment: * *low hysteresis is decisive for multi-state operation. I'm a bit surprised that even under current control quite big hysteresis occurs under some conditions, e.g. fig. 2c,d. Any idea why, and how to improve that?*

Response: We ascribe the observed hysteresis during the current controlled gating to the kinetics of proton diffusion in the channel layer. Based on the diffusion coefficient we measured in Figure S13, which drops quickly with the intercalation of proton, the time needed for the proton gradient to equilibrate between the interface of Nafion/ WO_3 and the WO_3 bulk is significantly longer with higher hydrogen content. Therefore, with the same sampling period, we had to acquire the non-equilibrated states at higher H content region, leading to asymmetric behavior.

To further reduce the asymmetry, miniaturizing the dimension of the channel material, including reducing thickness and channel length will be most promising. This will significantly reduce the equilibration time needed for the proton to diffuse, revealing the true steady-state conductance. Also, this can reduce the current level needed to achieve similar effect of gating, which is usually accompanied by increased Faradic efficiency. In addition, when operated in smaller H exchange range, the asymmetry can be better than operating with high on/off ratio. For example, the data below shows the low hysteresis when only operated in the extreme case with 1 potentiation/ 1 depression for 100 cycles. Also, as seen in Figure 1d, when the pulse height is 200nA, the inserted hydrogen amount during each pulse is less, and the equilibration is faster, thus giving a more symmetric potentiation/depression behavior even with 1000 pulses.

Figure R1(also S16). a) Conductance (G) and b) conductance change (ΔG) with alternating 1 potentiation and 1 depression operation for 100 cycles. Reproducible switching behavior with low hysteresis is evident.

We have added this comment in the main text on Page 11.

Comment: * *According to the reported results, current gating is preferred as operation mode over voltage gating. Please comment if this would lead to any significant complications for*

construction of real devices comprising many such synapse cells (current is more difficult to control than voltage), or how big the chance is to get good performance also in voltage gating

Response: We thank the reviewer for noting the difficulty of constructing constant current source. Indeed, constant current source is less common. In literatures, people have successfully fabricated current source that can supply fast pulse with discrete PFET and NFET. (Tang, J. *et al.* in *2018 IEEE International Electron Devices Meeting (IEDM)*. 13.11.11-13.11.14) This circuit has been used to gate the ECRAM in their demonstration. Therefore, it indeed will need extra circuit elements to achieve the goal, but we believe it is still feasible.

Fig. 7. ECRAM unit cell design for high-speed programming, in which discrete PFET and NFET serve as current source to ECRAM for positive (potentiation) and negative (depression) weight update, respectively.

Figure R2. Realization of current source demonstrated in the literature with fast pulsing capability. (Tang, J. *et al.* in *2018 IEEE International Electron Devices Meeting (IEDM)*. 13.11.11-13.11.14)

We have also performed the test under constant voltage gating. The result and related discussion have been shown in Figure S12. With proper selection of the specific gating voltage, good performance can also be achieved. The only complication during this process is the lack of theoretical guidance to choose the proper gating voltage to get the symmetrical behavior, which could be resolved by calibration curves.

Comment: * *it would be good to comment on the scatter in conductance between a larger number of nominally identical cells (i.e. how reliably can the number of cells be scaled up)*

Response: We believe that reproducibility is a major advantage of our switching mechanism vs the conductive filament mechanism. The main cause of lack of reproducibility of the conducting filament devices is the stochastic nature of the filament formation. In contrast, the protonation mechanism is deterministic and charge controlled. The conductivity of our channel is defined by a bulk material property, which is the hydrogen amount in H_xWO_3 . This deterministic approach that uniformly protonates or deprotonates the channel material should give a much more reproducible behavior. Quantification of this as a function of device geometry and material type

will be the scope of our future work. We have added a discussion about this point on Page 5 of the main text.

Comment: * *the cells need humidity in the Nafion layer, and oxygen in- and/or hydrogen out-diffusion must be avoided. Authors report some preliminary encapsulation tests, but longer times than in fig S14 should be reported*

Response: We agree that the encapsulation, and the resulting retention and endurance should be more thoroughly characterized. Currently for our proof-of-concept device, the encapsulation approach is far from being ideal, and rather rudimentary. So any data that we collect with this approach will not represent the true potential of this device for endurance and retention. But we do have longer testing data of the present encapsulated device, that was monitored for 21 hours after a sequence of potentiation cycling as shown in Figure R4 (Figure S15a) below.

Figure R3. Retention of potentiated state of an encapsulated device over 21 hours. Before $t=0$, the conductance was 30 nS. At $t=0$, 10 potentiation pulses doubled the conductance of the channel to 60 nS. The gating circuit was opened and the channel conductance was continuously monitored for 21 hours. The final conductance was 47 nS. To be noted, the encapsulation applied here is still far from ideal, but the result is promising and suggests good retention.

This demonstration is now included in Supplement Figure S15a.

Comment: * *a related issue: Nafion is an acidic polymer. Any long-term degradation in contact with palladium hydride (the hydrogen reservoir) by releasing hydrogen from proton+hydride ions, and incorporation of Pd^{2+} into Nafion for charge compensation?*

Response: Although known and named as Palladium hydride, this is not an ionic hydride. Hydrogen exists as H^0 in Pd metal, forming a solid solution (or an alloy). There is no negatively

charged H⁻ species, so the reaction with H⁺ in Nafion is not a concern. Palladium as a noble metal, has high redox potential for Pd²⁺ to form. The reaction of Pd dissolution into Nafion is not very likely within the potential window in which we are operating, which is well below 0.5 V.

Reviewer #2 (Remarks to the Author):

In this manuscript the authors present a very complete work on the development of three-terminal synapses based on proton intercalation in inorganic materials. The work is very complete, as it provides clear information about the fabrication of the devices, complete morphological and electrical characterization, and atomistic modelling. The electrical characteristics measured are impressive, specially the linearity achieved and the controllability of the potentiation and depression processes. The conductance modulation mechanism is clearly identified using “in operando” chemical measurements. Overall, the work provides important insights in its field of research, and the reliability of the findings is higher than in most reports in this field due to the multiple and high quality techniques used. In my opinion, this work is above the level of most others in the field, and for that reason I would like to recommend its publications in Nature Communications.

The only weakness that I see, which the authors should definitely further discuss, is the integration capability of this work:

Comment: *1- First of all, they use devices with channels of 100 um (is that the length or the width?). These devices are very big. Did the authors try to do smaller devices? This is highly recommended in order to observe if the findings also apply to miniaturized devices.*

Response: We agree that the scaling behavior of this device should be thoroughly characterized. The scope of this current paper is to demonstrate the concept and working mechanism of the material and device system. Our present devices were patterned by shadow masks. This limits the size and geometry of the device. However, we are currently working on downscaling the protonic electrochemical synapses with a much smaller footprint by photolithography and an improved layout. This extensive work will be reported in a future publication, and we believe it is out of the scope of our current paper. We hope this is acceptable to the reviewer.

Comment: *2- Moreover, as the authors wrote the word “computing” in the title of their abstract and because the introduction of their paper is discussing about “in memory computing”, the authors should discuss what is the potential of their device for computation, which in its current form seems to be unclear to me. The authors should clarify which kind of computation may be done with their device and the path towards such implementation. I do not think it is necessary they demonstrate such computation, but they may reference some works that did such things.*

Response: We thank the reviewer for clarifying the importance of technological background. We have modified the introduction to include specific examples of the applications. Please find the related revision on Page 3 of the main text.

Comment: *3 – The authors should also indicate which is the advantage and disadvantage of their devices compared to two-terminal resistive switching devices. I think two-terminal resistive switching devices are the most commonly used in most companies, and therefore it seems to me*

that they have most potential to end up being the hardware for such “in memory computing”. Therefore, from a technological point of view, the authors should clarify the potential of their work.

Response: We have modified the introduction part to clarify the potential of our work. Please find related revisions on Page 3 and Page 5 of the main text.

Comment: *The work is really good, I really enjoyed reading it, but the authors should provide more practical information about the technological potential of their work. I also recommend to remove the word “computing” from the title; some readers may get too excited when seeing the title and later disappointed when reading the manuscript, as there is not computing in this manuscript. In the future, the authors may try to associate different devices to solve a complex problem. They may even indicate such thing in the conclusions to encourage the community to work in such application.*

Response: We have changed our title to address this concern. The new title reads as follows:

Protonic Solid-State Electrochemical Synapse for Physical Neural Networks

We also modified the conclusion section to state the potential of our device on Page 19 of the manuscript.

Reviewer #3 (Remarks to the Author):

Authors in this work demonstrate a three terminal device based on proton migration. The device shows a great reversibility for analog signal processing under current pulses. They have also performed impressive and careful materials studies both experimentally and theoretically to conclusively reveal the switching mechanism. It is a very interesting and timely work. I strongly recommend it for publication.

I do have some minor comments for the authors to consider.

Comment: 1. *The authors obtained good XAS and XPS results to make comparisons between WO_3 and H_xWO_3 , but the exact ratio of O to W of the as-deposited film was not revealed. It would be great if the stoichiometric ratio of O to W is also given in the manuscript. Did author fabricate the WO_x film with a non-stoichiometric ratio? How would that affect the performance of the device? How to eliminate the effect of oxygen vacancies migration in WO_x channel?*

Response: Upon deposition by sputtering, we have annealed the films in ambient air. Therefore, we expect them to be (nearly) fully oxidized, WO_3 . Based on the XPS spectrum of W4f region and oxidation state analysis below, we conclude that the O:W ratio is at least 2.98.

We used the W4f spectrum analysis to estimate the non-stoichiometry. The W 4 $f_{7/2}$ peak was deconvoluted into two peaks. The main peak was assigned to be W^{6+} and a very small shoulder peak was assigned to the reduced W^{5+} (for instance due to oxygen vacancies in the as-prepared state). Based on the peak position, these two peaks are ascribed to W^{6+} and W^{5+} , respectively. The relative percentage of these two peaks are 95.5% vs 4.5%. This gives an O:W ratio as 2.98. So, we use the notation of WO_3 instead of WO_x throughout our manuscript.

If the O:W ratio were to be low, the initial conductance of the pristine channel will be higher, as a result of mixed oxidation state of W. But the device can still switch, just with a smaller G_{\max}/G_{\min} ratio.

As to the oxygen vacancy migration in WO_3 channel, we do not expect oxygen to be able to move at room temperature because the field crated between the Source-Drain electrodes is very small. We are using only 0.1 V as reading bias, which is too small to induce oxygen vacancy migration at room temperature.

Comment: 2. The WO_3 film was annealed in air at 450 °C for 1 h. Please comment on the concentration and effects of nitrogen in the annealed film. In addition, 450C is higher than 400C, which means this process is not 100% fab compatible yet. Would 400C annealing for a longer time achieve the same effect?

Response: We examined the N1s region of XPS survey spectrum and didn't find any inclusion of N. N_2 is still considered inert at 450 °C toward WO_3 .

Figure R4. a) N 1s region of XPS survey spectrum obtained from 450°C- WO_3 sample, with no observable N signal. b) Potentiation/depression behavior of 400 °C- WO_3 device. The switching behavior and symmetry is also excellent.

Regarding lower fabrication temperature, we have annealed the WO_3 film at 400 °C for 1h. Although the film is still amorphous, as examined by XRD, the device can deliver the desired switching behavior. We have added the potentiation/depression data of the WO_3 film annealed at 400 °C as Supplementary Figure S5, and shown above in Figure R4(b). We used the 450 °C sample in our main text due to the well-defined crystal structure for the ease of analysis.

As shown above, crystallization of WO_3 is not necessary to induce the proton induced resistive switching behavior. However, if crystallized WO_3 is more desirable, it is possible to obtain crystallized structures at reduced temperatures below 400 °C. Cold sintering, as an example, can accelerate the crystallization of material with lower temperatures (Guo, J. et al. Angew. Chem., Int. Ed. 55, 11457-11461 (2016)). If needed, these modern techniques are available to reduce the processing temperature of such materials and devices, but this also warrants future work to explore.

Comment: 3. Please comment on how to control concentration of hydrogen in Pd, nafion and WO₃ during fabrication process? How do we know that H has been incorporated into the Pd films? Can phase/structure of PdH_x or Hydrogen forward scattering (HFS) help?

Response: The introduction of hydrogen into our device was through the chemisorption of H₂ by Pd. The amount can be controlled by the exposure time, hydrogen partial pressure and reaction temperature. Once H is in the Pd, Pd expands, so the XRD can help quantify on how much H is in Pd. Advanced characterization techniques, such as Hydrogen forward scattering or neutron scattering could also give quantitative information of hydrogen concentration.

In our experiments, we have used the open-circuit potential measurement to determine, *in situ*, if the hydrogen absorption has been realized. Before the introduction of hydrogen, the open circuit potential of our device will be defined by the O₂ potential adsorbed on Pd (close to 1.23 V vs. Standard Hydrogen Electrode), which has a positive potential vs. the WO₃/Au electrode. When Hydrogen gas is introduced, H incorporate into Pd to form PdH_x which significantly reduces the potential to nearly 0 V vs. SHE. This makes the open circuit potential of PdH_x negative with respect to WO₃/Au electrode. Based on this observation, we can know if H has been incorporated into Pd or not. We have added this observation into the Experimental methods section on Page 20 of the main text now.

Comment: 4. The relative humidity was 100% during the testing. Does it mean that the devices still somehow rely on water electrolysis? Would any change of the environment humidity affect the proton conductivity of the device? Would the device work well when the test chamber contains hydrogen only? What is the chamber pressure when the devices were tested?

Response: Lack of water hydrolysis is actually a key difference of our work vs. prior work. We do not rely on water electrolysis to supply protons, which is evidenced by the small gating voltage we are using here (<0.5V). To electrolyze the water, the minimum of 1.23 V will necessary to meet the thermodynamic requirement and an additional 0.5~0.7 V is needed to break the kinetic overpotential, even on the best electrocatalyst. So we can confidently rule out the possibility of water electrolysis here. We have added this new discussion on Page 11 of the manuscript.

With the open structure of our demo device, the humidity does have an influence on the proton conductivity of the electrolyte. This is why the reason for measuring in nearly 100% humidity was to maintain the optimum conductivity of the Nafion electrolyte. We indeed have measured the device in dry hydrogen only. There was no difference in terms of switching behavior, but there was an increase of gating voltage at the same gating current. This increase is a result of reduced electrolyte conductivity, which increases the internal iR loss. We have also tested an encapsulated device, which can be done in ambient air, as shown in Figure S6.

The testing chamber for the open structured device was at ambient pressure (~760 torr). The estimated gas composition is Ar (92%), hydrogen (5%), water (3%).

Comment: 5. In Figure 2b-d, the conductance updating property of the device at low conductance regime shows a better symmetry than that at middle and high conductance regime.

Why does the symmetry of the conductance updating become worse when the concentration of hydrogen increases? Why there is still an asymmetrical updating in low conductance regime? Since the symmetry of conductance updating with 200 pulses looks better than that with 40 pulses and 100 pulses in low conductance regime, would the symmetry be affected by pulse number?

Response: The reason for the observed hysteresis is more obvious in the high conductance region which has a higher hydrogen concentration. We ascribe the observed hysteresis during the current controlled gating to the kinetics of proton diffusion in the channel layer. The diffusion coefficient we measured in Figure S13, drops quickly with the intercalation of protons. Therefore, the time needed for the proton gradient to equilibrate between the interface of Nafion/WO₃ and the WO₃ bulk is significantly longer with higher hydrogen content. We keep the same sampling period for all protonation regimes, and while this has been sufficient to obtain equilibrated data for the low protonation regime, it is too short for the higher protonation regimes (i.e. the equilibration time is longer than our sampling period). Therefore, we ended up having to acquire the non-equilibrated states at higher protonation regimes, and this leads to an asymmetric behavior.

For the low conductance regime, symmetry is better but some asymmetry is still present, because the amount of H exchanged at each pulse is still considered relatively high, due to the limitation of our current instrumentation. To further reduce the asymmetry, miniaturizing the dimensions of the channel material, reducing the pulse current, and shortening the pulse length will be the most promising. This will significantly reduce the equilibration time needed for the proton to diffuse, revealing the true steady-state conductance. Also, this can reduce the current level needed to achieve similar effect of gating, which is usually accompanied by increased Faradaic efficiency.

We attempt to clarify the symmetry comparison here. The symmetry will be affected by pulse number, but is the opposite than what the reviewer described above. Smaller pulse number means smaller deviation from the equilibrated state of the material, which results in better symmetry. An extreme case is when only 1 potentiation and 1 depression is applied, in which case, we can obtain nearly perfect symmetry, as shown in Figure R5 (also Figure S16) below. With the increase of pulse number, the diffusion factor mentioned above will result in gradual decrease of ΔG per pulse at higher pulse number, resulting in higher asymmetry.

Figure R5 (also S16). a) Conductance (G) and b) conductance change (ΔG) with alternating 1 potentiation and 1 depression operation for 100 cycles. Symmetrical ΔG of potentiation and depression is observed.

Comment: 6. *In Figure 3b, is there a plausible explanation for that the high resistive state of the device becomes more insulative and low resistive state of the device becomes more conductive after large amounts of cycling?*

Response: First, we want to note the change of G_{\max} and G_{\min} are relatively small in Figure 3b. This trend is not universal to all devices. This phenomenon could be a result of increased Faradaic efficiency during the cycling when the irreversible side reactions are gradually consumed. So, the same level of current can lead to slightly higher amount of actual hydrogen intercalation.

Comment: 7. *on Page 7, there is a typo, “As demonstrated in Figure 1a, over seven orders ...” should be Figure 2a.*

Response: We have made the correction (on Page 8 now) and examined the full manuscript again to make sure the consistency.

Comment: 8. *on Page 9 about Figure 2a “...after 100 positive gating pulses, the conductance of the channel increased from 1 μS to 22 μS , ...” it's a bit confusing here, what is the pulsing sequence, from 100 down counting to 1 in Figure 2b? please use arrows to label them in the figure.*

Response: We have added arrows indicating the direction of operation in Figure 2b-d for clarification.

Comment: 9. *The specific pulse height-width that was used to achieve each conductance regime (low, medium and high) should be clearly stated in the caption of Figure 2 or the main text.*

Response: We have added this information to the caption of Figure 2 to indicate the pulses needed to modulate the conductance to the desired level.

Comment: 10. *Endurance to the level of million has not been demonstrated (and not really expected for this paper as well). Is that due to testing time or failure? Please comment on possible failure mode?*

Response: We do have limitations at the current stage of development for cycling test. With the present pulse height/width, our testing protocol requires 1s for each cycle for stable conductance readout, 1 million cycles with 100p/100d will require 6 years for test. We're in the process of setting up to use nA and ns pulse regime, and examining the endurance in that regime will be more feasible. Regarding the failure mode, the devices do fail after prolonged operation. The most common failure mode is the increase of internal resistance as a result of Nafion electrolyte degradation, due to the loss of water content to the environment. A better electrolyte choice that do not rely on hydration to conduct proton is desired, and we're also working on this. Another major reason for the failure of encapsulated device is the depletion of active hydrogen in the

system. As we discussed in *Supplementary Discussion 2*, the ingress of oxygen, diffusion of hydrogen, and less than 100% Faradic efficiency will all contribute to this aspect. However, our present encapsulation is far from ideal, and this behavior does not represent the true potential of the device. The last failure mode is the degradation of electrode material. After the hydrogen is absorbed by Pd, there will be a relatively large volume expansion (as PdH_x is an alloying reaction, not intercalation reaction), upon repeated cycling, Pd layer may detach from Nafion surface. This requires improved adhesion layer at Nafion/Pd interface or other material choice as hydrogen reservoir other than Pd metal.

We have expanded the discussion to reflect on the potential failure modes of these devices on Page 12-13 in the main text.

Comment: *11. As the author mentioned, “When the external circuit is cut off (gate is open circuit), the electronic insulating nature of the electrolyte prevents the back flow of electrons, and consequently that of protons too.” Therefore, a two-terminal threshold volatile switch was used to provide both selector and threshold functions for programming (e.g. NatMat 16, 396, 2017), without which an array of such devices may only be programmed column-wise or row-wise (not randomly and individually programming a device in the array during learning). Will the device studied here also need such as threshold switch on the gate?*

Response: For the current device configuration, there is a non-zero open circuit potential (OCP) change after the gating, so the selector is needed to program individual synapse in an array, as demonstrated by recent publication in *Science* **364**, 570-574 (2019), and to avoid cross-talk of individual synapses when not gating. We have added notes on Page 12 in the manuscript to reflect this point.

If the device possesses OCP = 0 V or the change of OCP after gating remain the same, there is no need for the two-terminal selector to program each individual synapse. The small OCP change after gating has been successfully demonstrated in our work as shown in Figure 2a, for the $x=0.6\sim 1$ region. But it is still far from desirable and the corresponding conductivity change in this region is relatively small. Further work is on-going in our lab to achieve the true zero-OCP change during the gating.

Comment: *12. A constant current source (compared to a voltage source) is usually not preferred in real circuits due to various reasons.*

Response: We are aware of this challenge and have performed the test under constant voltage gating, too. The result and related discussion are shown in Figure S12. With proper selection of the specific gating voltage, good performance can also be achieved with our device.

In the literature, researchers have successfully fabricated current source that can supply fast pulse with discrete PFET and NFET. (Tang, J. *et al.* in *2018 IEEE International Electron Devices Meeting (IEDM)*. 13.11.11-13.11.14) This circuit has been used to gate the ECRAM in their demonstration. Therefore, it indeed will need extra circuit elements to achieve the goal, but we believe it is still reasonable.

Fig. 7. ECRAM unit cell design for high-speed programming, in which discrete PFET and NFET serve as current source to ECRAM for positive (potentiation) and negative (depression) weight update, respectively.

Figure R6. Realization of current source demonstrated in literature with fast pulsing capability. (Tang, J. *et al.* in *2018 IEEE International Electron Devices Meeting (IEDM)*, 13.11.11-13.11.14)

REVIEWERS' COMMENTS:

Reviewer #1 (Remarks to the Author):

the revision answers my questions adequately, paper can be published now

Reviewer #2 (Remarks to the Author):

The authors have made a good revision of my previous comments, and I think the manuscript reaches the level of other papers published in Nature Communications in this field of research. I believe publishing this manuscript is a good choice.

Just one minor point. I recommended the authors to remove the word "computing" from the title and they did. But they replaced by "neural network". I think this manuscript demonstrates in a very clear way that the devices exhibit some synaptic behaviors, but they do not go beyond that. I think the authors should still modify the title and remove any work that points to association of devices (such as computing or neural network), otherwise the readers is going to be confused and overall gives the impression that the authors try to oversell their work. The work is already excellent in its field, such overselling is not necessary. So, better to change the title and make it more descriptive of the content.

Other than this, the paper is excellent. Congratulations to the authors for this excellent achievement.

Reviewer #3 (Remarks to the Author):

The authors have addressed my questions satisfactorily in the revision, which is recommended for publication as is now.

REVIEWERS' COMMENTS:

Reviewer #1 (Remarks to the Author):

the revision answers my questions adequately, paper can be published now

Response: We are pleased to find the reviewer is satisfied with our revision and thank the reviewer for the support.

Reviewer #2 (Remarks to the Author):

The authors have made a good revision of my previous comments, and I think the manuscript reaches the level of other papers published in Nature Communications in this field of research. I believe publishing this manuscript is a good choice.

Just one minor point. I recommended the authors to remove the word “computing” form the title and they did. But they replaced by “neural network”. I think this manuscript demonstrates in a very clear way that the devices exhibit some synaptic behaviors, but they do not go beyond that. I think the authors should still modify the title and remove any work that points to association of devices (such as computing or neural network), otherwise the readers is going to be confused and overall gives the impression that the authors try to oversell their work. The work is already excellent in its field, such overselling is not necessary. So, better to change the title and make it more descriptive of the content.

Other than this, the paper is excellent. Congratulations to the authors for this excellent achievement.

Response: We appreciate the review’s support of our paper, constructive review, and suggestion to modify the title again. We revised our original title per this reviewer’s comment, and included the “physical neural network” in the title to indicate the targeted application field of our artificial synapse. This clarification of potential application is also in line with the reviewers’ recommendation in the first round of review, *“The authors should clarify which kind of computation may be done with their device and the path towards such implementation.”*. We believe this will help to distinguish our study from the studies on the biological synapses, and also from biological applications of artificial synapses (neuroprosthetics). The presence of “physical neural networks” helps to better reach the targeted readership of this article. We hope you can support our preference in keeping the title as is now.

Reviewer #3 (Remarks to the Author):

The authors have addressed my questions satisfactorily in the revision, which is recommended for publication as is now.

Response: We are pleased to find the reviewer is satisfied with our revision and thank the reviewer for the support.